# Near-Ambient Pressure XPS and MS Study of CO Oxidation over Model Pd-Au/HOPG Catalysts: The Effect of the Metal Ratio

**DOI:** 10.3390/nano11123292

**Published:** 2021-12-04

**Authors:** Andrey V. Bukhtiyarov, Igor P. Prosvirin, Maxim A. Panafidin, Alexey Yu. Fedorov, Alexander Yu. Klyushin, Axel Knop-Gericke, Yan V. Zubavichus, Valery I. Bukhtiyarov

**Affiliations:** 1Synchrotron Radiation Facility SKIF, Boreskov Institute of Catalysis SB RAS, 630559 Kol’tsovo, Russia; mpanafidin@catalysis.ru (M.A.P.); yvz@catalysis.ru (Y.V.Z.); 2Department of Physicochemical Techniques, Boreskov Institute of Catalysis SB RAS, 630090 Novosibirsk, Russia; prosvirin@catalysis.ru (I.P.P.); afedorov@catalysis.ru (A.Y.F.); vib@catalysis.ru (V.I.B.); 3Department of Natural Sciences, Novosibirsk State University, 630090 Novosibirsk, Russia; 4Inorganic Chemistry Department, Fritz Haber Institute of the Max Planck Society, 14195 Berlin, Germany; klyushin@fhi-berlin.mpg.de (A.Y.K.); knop@fhi-berlin.mpg.de (A.K.-G.); 5BESSY II, Helmholtz Center for Materials and Energy, 12489 Berlin, Germany; 6Department of Heterogeneous Reactions, Max Planck Institute for Chemical Energy Conversion, 45470 Mülheim an der Ruhr, Germany

**Keywords:** NAP XPS, bimetallic Pd-Au nanoparticles, CO oxidation, adsorption-induced segregation

## Abstract

In this study, the dependence of the catalytic activity of highly oriented pyrolytic graphite (HOPG)-supported bimetallic Pd-Au catalysts towards the CO oxidation based on the Pd/Au atomic ratio was investigated. The activities of two model catalysts differing from each other in the initial Pd/Au atomic ratios appeared as distinctly different in terms of their ignition temperatures. More specifically, the PdAu-2 sample with a lower Pd/Au surface ratio (~0.75) was already active at temperatures less than 150 °C, while the PdAu-1 sample with a higher Pd/Au surface ratio (~1.0) became active only at temperatures above 200 °C. NAP XPS revealed that the exposure of the catalysts to a reaction mixture at RT induces the palladium surface segregation accompanied by an enrichment of the near-surface regions of the two-component Pd-Au alloy nanoparticles with Pd due to adsorption of CO on palladium atoms. The segregation extent depends on the initial Pd/Au surface ratio. The difference in activity between these two catalysts is determined by the presence or higher concentration of specific active Pd sites on the surface of bimetallic particles, i.e., by the ensemble effect. Upon cooling the sample down to room temperature, the reverse redistribution of the atomic composition within near-surface regions occurs, which switches the catalyst back into inactive state. This observation strongly suggests that the optimum active sites emerge under reaction conditions exclusively, involving both high temperature and a reactive atmosphere.

## 1. Introduction

The ability of supported bimetallic catalysts to outperform their monometallic counterparts has induced a great deal of interest from the scientific community in these systems as reported in numerous recent papers [1,2,3,4,5,6,7,8,9,10]. In particular, the PdAu system is among most studied due to its enhanced catalytic properties in a number of reactions, such as vinyl acetate synthesis [11], low-temperature CO oxidation [12,13], NO reduction [14], direct formation of hydrogen peroxide from an H_2_ + O_2_ mixture [15] and others. However, despite a large number of investigations on Pd-Au bimetallic catalysts being published in the last decade, the structure of catalytic active sites even in this prototypic system remains elusive, since the actual ratio of the metals on the surface of a working catalyst is not determined unambiguously by the amount of initially introduced metals, but is also strongly affected by the calcination temperature. Moreover, a redistribution of the metals in a particle can occur directly during the catalytic reaction, under the influence of the temperature and reaction mixture [6,7,8,9,10]. To address the problem, *operando* studies have to be performed. However, industrially applicable catalysts are characterized by severe limitations, such as the objects being probed by surface-sensitive techniques due to the low bulk concentration of the active component and the irregularly shaped flakes of the high-specific-area support. Fundamental aspects of the advanced catalyst design are better elaborated with simpler model catalysts. In particular, metal nanoparticles deposited onto the atomically smooth highly oriented pyrolytic graphite (HOPG) support as a monolayer adequately mimic essential details of the catalytic performance of real carbon-supported nanostructured catalysts and can be efficiently characterized by standard surface science-oriented instrumental techniques at the same time [16,17]. In recently published papers [18,19], model alloyed Pd-Au nanoparticles deposited as a monolayer onto the HOPG support were tested towards the oxidation of carbon monoxide using a set of techniques including near ambient pressure X-ray photoelectron spectroscopy (NAP XPS) and mass spectrometry (MS). The catalytic activity of the samples was ignited at temperatures higher than 150 °C. An exposure of the bimetallic nanoparticles to CO under reaction conditions has been demonstrated to induce segregation that manifests itself as an enrichment of the surface with Pd, since the emergence of favorable Pd–CO-bound states mitigates the energy costs required for the intraparticle atom redistribution. It has been shown that heating of a sample under reaction conditions above 150 °C gives rise to the decomposition of Pd–CO states due to CO desorption followed by oxidation, which simultaneously results in Pd-Au surface alloy reformation. Thus, it has been clearly demonstrated that it is the alloyed surface that is responsible for CO oxidation. This conclusion is in line with Goodman’s mechanism [13,18], which suggests that gold atoms are responsible for the CO adsorption, whereas contiguous Pd sites are responsible for the O_2_ dissociation and a further O_ads_ spillover to Au or to isolated Pd sites followed by the oxidation of the activated adsorbed CO. It should be noted that this mechanism implies that the oxidation of carbon monoxide is initiated at moderately low temperatures when a required concentration of adsorbed CO molecules is reached due to rather weak bonding of CO to Au. The higher extent of surface segregation of palladium atoms induced by adsorption of carbon monoxide molecules gives rise to a disturbance of the alloy structure. As such, the surface of the nanoparticle becomes fully inhibited and the catalysts demonstrate essentially zero activity. We suggested that this indeed occurred with the samples studied in [18]; the starting surface atomic ratios Pd/Au of samples exposed to the CO + O_2_ atmosphere were quite high at ~1.4 and 3.5 for the model bimetallic Pd-Au/HOPG catalysts under study. In such cases, another carbon monoxide oxidation mechanism can come into force. Indeed, the significant deficit of the topmost surface Au atoms makes the system more similar to single-component palladium nanoparticles. This is indeed the case, since the Pd-Au/HOPG catalysts were inactive towards the CO oxidation in the low-temperature mode, whereby the activity ignited only above 150 °C. Finally, it was suggested that model alloy Pd-Au/HOPG samples characterized by a lower Pd/Au ratio should demonstrate greater activity in the low-temperature mode below 150 °C. A dedicated comparative study of such catalysts with different metal ratios could help to obtain more reliable data concerning the nature of the bimetallic catalytic system and the mechanism of its activation. This information would extend the utility of Pd-Au/HOPG as a model catalytic system. Furthermore, it could be used as a rational basis for the preparation of efficient Pd-Au bimetallic or even Pd single-site catalysts to be applied in various industrially relevant reactions.

In this paper, we have compared the results obtained for two Pd-Au/HOPG model bimetallic catalysts with different Pd/Au ratios on the particle surfaces. The key goal of this investigation was to obtain a correlation between the catalytic activity of those model catalysts towards the low-temperature CO oxidation and the surface Pd/Au atomic ratio under reaction conditions. We anticipated that these two catalysts would demonstrate distinct proneness to adsorption-induced segregation, which would shed light on the mechanism underlying this effect and extend the synthetic capabilities to tune the structure and properties of catalysts. The electronic properties of the metals constituting bimetallic palladium–gold particles and their concentration depth profiles were systematically elucidated, along with their transformation induced by a reactive environment typical of CO oxidation reaction, for different reaction conditions and initial Pd/Au ratios.

## 2. Materials and Methods

Samples preparation and characterization were performed in a photoelectron spectrometer manufactured by SPECS (Berlin, Germany) equipped with a PHOIBOS-150-MCD-9 hemispherical analyzer, a FOCUS-500 ellipsoidal monochromator and an XR 50 M X-ray source with a double Al/Ag anode. Commercially available HOPG (square plates 7 × 7 mm^2^, approx. 1 mm thickness; SPI supplies (West Chester, PA, USA), Grade SPI-2) was used as a support for bimetallic Pd-Au catalysts. For the catalyst preparation, the Pd and Au foils of a high purity grade (99.99%) were used. Small pieces of the foils were placed inside a tantalum crucible and evaporated using an EFM3 electron beam evaporator (Scienta Omicron AB, Uppsala, Sweden). Deposition conditions were as follows: accelerating voltages were ~900 V for Pd and 800 V for Au, while the thermoemission current was ~15 mA for both metals. The amount of metal deposited varied based on the duration of the deposition and was controlled with XPS. Two samples with alloyed bimetallic Pd-Au nanoparticles supported on defect surface of HOPG with different mean particle sizes and Pd/Au ratios were prepared by thermal vacuum deposition of Pd onto an Au/HOPG matrix with a subsequent step of high-temperature treatment at 400 °C in an ultra-high vacuum (UHV) in order to achieve complete intermixing of Pd and Au atoms within bimetallic nanoparticles [18,19,20,21]. For the preparation of the monometallic Au/HOPG samples, which were then subsequently used as templates for palladium deposition, the earlier reported three-step procedure was used [21]. First, this included the introduction of surface defects onto the HOPG surface by a mild Ar^+^ etching (t = 3–4 s, p(Ar) = 3 × 10^−6^ mbar and accelerating voltage of 0.5 kV). The first step was followed by the gold deposition using an Omicron EFM3 electron beam evaporator (Scienta Omicron AB, Uppsala, Sweden). The third and final step included defect annealing and particle stabilization at 300 °C under UHV conditions. The samples were investigated using scanning tunneling microscopy (STM) and X-ray photoelectron spectroscopy (XPS) at all stages of the preparation procedure.

STM measurements of the prepared Pd-Au/HOPG samples were carried out using an UHV 7000 VT microscope (RHK Technology, Troy, MI, USA) operating in the constant current mode with cut Pt–Ir tips. The Si(111) single crystal with the 7 × 7 surface reconstruction and clean HOPG were used as references for the scanner calibration. The STM images were processed using the ParticlesNN web service [22,23,24]. In order to convert initial STM images to digitized matrix files appropriate for input to the web service, WSxM 5.0 software was used [25]. The mean particle size (<*d*>) was determined according to the following equation:$$\langle d\rangle =\frac{\sum_{i}\left(d_{i}\times N_{i}\right)}{\sum_{i}\left(N_{i}\right)}$$
where *N_i_* denotes the number of particles with a diameter *d_i_* (the summation is performed over all particles recognized in an image, typically *i* > 2000).

NAP XPS measurements were accomplished using a dedicated NAP XPS unit at the ISISS beamline at BESSY II/HZB (Berlin, Germany) [26]. The Pd-Au/HOPG and Pd/HOPG model nanostructured catalysts were sandwiched between the backplate and lid (having a 6 mm hole drilled in the center) made of stainless steel. Then, the samples were fixed onto a special sample holder of sapphire ceramics. The sample back heating was achieved with an infrared laser, whereas a K-type thermocouple was applied to control the temperature throughout the measurements. The mixture of carbon monoxide and oxygen taken at a proportion of 2:1 (CO/O_2_) was fed into the high-pressure cell. The pressure within the cell was slowly increased to 0.25 mbar using MFC mass-flow controllers (Bronkhorst, Ruurlo, The Netherlands). All samples were annealed at 400 °C in UHV before the CO oxidation tests in order to remove all contamination from the surface and to recover the alloyed structure disturbed by the storage in air. The experiments under CO oxidation reaction conditions were performed as follows: a CO + O_2_ flow (CO/O_2_ = 2:1; P_total_ = 0.25 mbar) at room temperature (RT) was fed into the gas cell, then the samples were heated step-by-step up to 250 °C (RT–150 °C–200 °C–250 °C) and then cooled down to RT in the gas mixture. A Prisma mass spectrometer from Pfeiffer attached to the reaction cell via the process line was used to analyze the gas-phase composition above the sample. For the purpose of depth profiling, all XPS lines from gold, palladium and carbon core levels (Au4f, Pd3d and C1s) were measured at several different incident photon energies to provide photoelectron kinetic energies of 300, 450 and 600 eV and to guarantee identical effective probing depths. To determine positions of the peaks in the Au4f and Pd3d spectra, XPS peaks were calibrated against the C1s spectra of the HOPG support (BE = 284.5 eV) taken at the same primary excitation energies as Au4f and Pd3d. The base pressure inside the spectrometer did not exceed 1 × 10^−9^ mbar for all experiments. Spectral analysis and data processing were performed with XPS Peak 4.1 software [27].

For the quantitative analysis, the integral intensities of Au4f, Pd3d and C1s lines were corrected using the ionization cross-section data taken from Yeh and Lindau [28] and normalized to the ring current and incident photon flux. To estimate the depths of analysis corresponding to different kinetic energies of 300, 450 and 600 eV, we used theoretically calculated apparent photoelectron inelastic mean free paths (IMFPs) for gold and palladium pure metal foils at 5.9, 7.6 and 9.2 Å, respectively [29]. For the Pd3d peak fitting, the Au4d_5/2_ contribution was subtracted to account for their overlap. There are two possible ways to take into account the Au4d_5/2_ contribution correctly. The first one is as described in [30], where best-fit parameters of the Pd3d_3/2_ component are applied to fit the Pd3d_5/2_ component using the theoretical spin-orbit splitting. The resultant residual intensity is, thus, ascribed to the Au4d_5/2_ line. The second way is based on fitting the Au4d_5/2_ component with parameters taken from the Au4d_3/2_ line, which does not overlap with the Pd3d line. Both these ways make sense and give quite similar results. In this work, the second one was used, i.e., the shape and intensity of the Au4d_5/2_ line were calculated from the less intense Au4d_3/2_ peak using the theoretical value of 2:3 for the branching ratio of the spin-orbit doublet components.

## 3. Results and Discussion

For the present study, two bimetallic alloyed Pd-Au/HOPG samples were selected. STM images of the prepared PdAu-1 and PdAu-2 catalysts are shown in Figure 1. A recent detailed study of Au nanoparticles supported on HOPG treated in argon [31] revealed that the gold nanoparticles predominantly had hemispherical or truncated hemispherical shapes. Thus, we can assume that the PdAu-1 and PdAu-2 samples prepared under very similar conditions should reveal similar shapes and mean sizes of the nanoparticles, i.e., characterized by hemispherical or truncated hemispherical shapes with average diameters of 4.7 nm and 4.6 nm, respectively, and narrow unimodal particle size distributions, as supported by experimental STM images.

Prior to in situ XPS experiments at different excitation photon energies (see experimental section), both samples were annealed in UHV at 400 °C to recover the alloyed structure disturbed by their storage in air [21]. Table 1 shows Au/C, Pd/C and Pd/Au atomic ratios calculated from the XP spectra measured at a photoelectron kinetic energy of 300 eV (surface localization) in UHV, which characterize the initial state of the catalyst surfaces. The samples reveal very similar values of the Pd/C atomic ratios but different Au contents, which imply successful preparation of two samples with different Pd/Au atomic ratios of ~1.0 for PdAu-1 and ~0.75 for PdAu-2 on the surfaces of bimetallic nanoparticles.

The NAP XPS spectra for the bimetallic model Pd-Au/HOPG nanocatalysts were measured under conditions of carbon monoxide oxidation with a total pressure of 0.25 mbar and CO-to-O_2_ ratio of 2:1_al_. Figure 2 shows concentrations of reaction product CO_2_ corrected for variations in the CO supply, as monitored by the mass-spectrometric signals of *m*/*z* = 44 together with Pd/Au atomic ratios calculated from Pd3d and Au4f XP spectra measured at different temperatures. The Pd/Au ratios presented in Figure 2 were determined based on X-ray photoelectron spectra acquired at a photoelectron kinetic energy of 300 eV (which corresponds to the surface localization). The Pd/Au atomic ratios increased from 1.0 to 1.45 and from 0.75 to 0.9 for the PdAu-1 and PdAu-2 samples under exposure to the CO + O_2_ reaction mixture at RT, respectively. The enrichment of the surface with Pd under CO oxidation conditions under similar conditions was previously revealed using near-ambient pressure X-ray photoelectron spectroscopy [18,19]. Moreover, rather moderate coverage is sufficient to make the segregation of Pd at faces of Pd-Au nanoparticles energetically feasible for nanoparticles similar in size to the ones experimentally studied here according to theoretical calculations based on density functional theory (DFT) and topological models for predicting CO adsorption energies [19]. It should be mentioned that the change in Pd/Au ratio is more pronounced for PdAu-1, which is characterized by a higher initial Pd/Au ratio than PdAu-2 (45% and 20%, respectively). Therefore, the difference in the Pd/Au ratios on the surface of bimetallic particles between those two samples becomes even greater as a result of CO adsorption-induced segregation under exposure to the reaction mixture at RT. From the MS data, one can see that the PdAu-1 sample demonstrates an essential absence of catalytic activity at T < 150 °C. Meanwhile, the intensity of the CO_2_ MS signal progressively increases above 150 °C and reaches a maximum at 250 °C. For thePdAu-2 catalyst, evidently enhanced CO oxidation activity can be observed. Indeed, the sample demonstrates measurable activity at temperatures less than 150 °C, which is lower by ~50 °C than for PdAu-1, showing activity at temperatures less than 200 °C. Reference experiments with pure HOPG showed strictly zero catalytic activity towards the oxidation of carbon monoxide under identical experimental conditions at all temperatures tested. For both samples, the Pd/Au atomic ratios decreased (Figure 2). This trend strongly implies that either the surface composition or structure of PdAu nanoparticles undergoes a drastic change under the action of the reaction mixture. If the sample is allowed to cool down to room temperature in the same atmosphere, the Pd/Au atomic ratio increases again, which means that the processes involved are completely reversible.

The shift of the activity ignition temperature to lower values for the sample with a lower Pd/Au surface atomic ratio was consistent with the assumptions made in earlier works published by Prof. Goodman’s group [13,18,32]. According to the proposed mechanism, the oxidation of carbon monoxide is expected to occur at a moderately low temperature when the required concentration of adsorbed CO molecules is reached due to weak interactions between CO and Au. For the PdAu-2 sample, the extent of the surface segregation of Pd atoms due to the CO adsorption and formation of Pd–CO bonds is moderate. Therefore, it can destroy the alloy structure and inhibit the metal surface completely, effectively deactivating the catalyst. The difference in the proneness of Pd atoms to segregation for those two samples could be explained by the initial Pd/Au ratios of the metals deposited onto the HOPG surface. Sitja et al. [33] recently reported on a molecular beam study of the CO adsorption on a regular array of PdAu clusters supported on alumina. They showed that an increase in the Au surface concentration leads to a decrease in the CO adsorption energy on PdAu clusters. They also have found that the ensemble effect is very important. On clusters of pure Pd, CO is adsorbed on hollow sites. With an increase in the gold surface concentration, CO first starts to adsorb on bridge sites and then on atop sites involving ensembles composed of 2 and 1 Pd atoms, respectively. Thus, the weaker proneness of Pd atoms to the CO adsorption-induced segregation observed for the PdAu-2 sample could be explained by the lower CO adsorption energy therein. Zhu et al. [34] attempted to optimize the surface d-charge of Pd-Au nanoalloy catalysts to enhance their catalytic activity towards the base-free oxidation of primary alcohols. The maximum d-charge gain was observed for single Pd atoms and dimers with 33–50 at.% Pd, which gave rise to a nearly 9-fold acceleration of the reaction in comparison with the monometallic Pd analogues. Thus, the catalytic activity and selectivity depend on mutual configurations of the Pd and Au atoms affecting the electronic properties of the nanoalloy. The surface Pd/Au ratio for pristine PdAu-2 (Table 1) is close to 0.75 (or ~43 at.% Pd), while it is ~1.0 (or 50%) for pristine PdAu-1 and increases even further to 0.92 (48%) and 1.5 (60%) for the PdAu-2 and PdAu-1 samples, respectively, due to CO-induced segregation under exposure to the reaction mixture. In the active state of both samples reached under the reaction conditions, the Pd/Au atomic ratios are close to those obtained for the initial state at 0.95 and 0.78 for PdAu-1 and PdAu-2, respectively. It could be suggested that independently of the segregation effects, the particles transform into the alloyed structure in the active state, which is directly responsible for catalysis. Therefore, analysis of the chemical composition evolution under reaction mixture depending on temperature is urgently required in order to understand the reasons for the different activities of these two samples.

Figure 3 shows the Au4f spectra for the PdAu-1 and PdAu-2 model bimetallic catalysts at RT (left panel), as well as plots of Au4f_7/2_ binding energies (BEs) (central panel) and Au-to-carbon atomic ratios (right panel) determined in situ in the atmosphere of carbon monoxide mixed with oxygen at a total pressure of 0.25 mbar and 2:1 reactant ratio (CO/O_2_) at different temperatures. Two states with different binding energies were observed for gold atoms. The main state at 84.0 eV for PdAu-1 could be attributed to metallic Au^0^ [18,21,35,36,37,38], while for PdAu-2, the XP spectra show a shift of the Au4f line to a lower binding energy of 83.8 eV, indicating that gold atoms are present as the Pd-Au alloy [18,19,20,21,39]. This difference in the BE values of Au4f_7/2_ components for the two Pd-Au samples could be explained by a more pronounced CO adsorption-induced Pd segregation for the PdAu-1 sample (with a higher Pd/Au ratio) and subsequent transformation of particles from a Pd-Au homogeneous alloy to core–shell structures with the Pd-enriched surface. According to literature data [21,40], the Au4f_7/2_ peak at higher BE (~85.0 eV) was assigned to Au anchored to carbon defect sites that were deliberately introduced into the support by a mild Ar^+^ sputtering of the HOPG surface. For the PdAu-1 sample, one can see that the Au4f_7/2_ BE shifts to lower values, whereas the Au-to-carbon atomic ratio increases as temperature increases. Cooling the sample back to room temperature in the gas mixture reverts the changes in spectral characteristics. The down shift of the Au4f_7/2_ component is an unambiguous manifestation of the solid solution emergence [18,41]. Meanwhile, the decrease in the Pd/Au atomic ratio upon heating is indicative of the Au surface segregation or diffusion of palladium atoms into the bulk of the bimetallic particles under conditions of carbon monoxide oxidation [21]. Under identical conditions, the analogous shifts in BE of Au4f_7/2_ for the PdAu-2 sample with a lower initial Pd/Au ratio is not so pronounced (~0.1 eV), which again demonstrates a lower extent of the CO-induced segregation of Pd atoms under exposure to the reaction mixture at RT, and correspondingly a lower degree of alloy decomposition.

Figure 4 shows the Pd3d XP spectra for the PdAu-1 and PdAu-2 model bimetallic catalysts measured in the in situ mode in the carbon monoxide/oxygen mixture (CO/O_2_ = 2:1, P_total_ = 0.25 mbar) at RT, 150 °C and 250 °C. The experimental spectra were analyzed based on reported deconvolution into individual components. The Pd encountered were assigned depending on the specific conditions of their occurrence [18,21,39,41,42,43,44]. Thus, the Pd3d_5/2_ component at a BE of 335.5 eV corresponded to metallic palladium [18,21,41,42], while the one at 335.1 eV was attributed to Pd in the PdAu alloy [21,39,43,44]. The Pd3d_5/2_ component with a BE of 335.9 eV was attributed to Pd–CO species emerging due to the adsorption of CO on the bridge or on-top Pd sites [18,43,44]. Finally, the Pd3d_5/2_ peak at 337.1 eV was assigned to small Pd clusters attached to some defect sites on the HOPG support [21,40,42].

All four species can be observed in Pd3d spectra of PdAu-1, while for PdAu-2 there is no metallic Pd species. Some of the Pd atoms are not involved in the solid solution formation and show up as the pure Pd metal component in the PdAu-1 sample, probably due to the excess of Pd. For both samples, the Pd–CO state (BE 336.0 eV) dominates on the surface at RT (Figure 4). At the same time, the PdAu alloy state is tiny (~10%) on the surface of PdAu-1, while for the PdAu-2 its fraction is significantly higher (~40%). These results correlate well with the respective differences in values of BE of Au4f_7/2_ peaks measured in the reaction mixture at RT (see Figure 3).

As discussed above, the PdAu-1 sample showed catalytic activity towards CO oxidation only at temperatures of 200 °C and higher, while for the PdAu-2 sample the catalytic activity was observed at temperatures lower than 150 °C. Previously, we demonstrated that the activity of Pd-Au/HOPG model catalysts is ignited by the decomposition of Pd-CO species and Pd-Au alloy reformation on the surface [18]. Importantly, for PdAu-2, there is an appreciable fraction of the Pd-Au alloy on the surface at RT in the reaction mixture. At temperatures above 150 °C (Figure 4 and Figure 5), when the PdAu-1 sample becomes active towards CO oxidation, the intensity of the Pd-CO state (BE = 336.0 eV) decreases simultaneously with an increase in the fraction of Pd in the Pd-Au alloy (BE = 335.1 eV). Under analogous conditions, for PdAu-2 sample that was already active at a lower temperature, only a slight decrease in the fraction of the Pd-CO state and an increase in that of Pd in the Pd-Au alloy can be observed. Finally, for active states of both samples at the maximum temperature (250 °C), the Pd component attributed to Pd in the Pd-Au alloy predominates on the surface (Figure 4 and Figure 5). Stromsheim et al. [43] have shown the occurrence of two distinct palladium states with Pd3d_5/2_ BEs close to each other (the difference is as small as ~0.2 eV) assigned to Pd-CO_ads_ and oxidized Pd-O species, respectively, depending on the experimental conditions, using a NAP XPS study combined with QMS to address CO oxidation over a Pd_3_Au(100) single-crystal face. Thus, the height of the Pd3d_5/2_ component at 336.0 eV that emerges after a temperature increase to 200 °C should not be due to the Pd-CO-adsorbed state since the Pd-CO_ads_ species are expected to be unstable at this temperature, and most likely it should be attributed to oxidized Pd-O species. The reverse atomic redistribution and surface composition change happens after the catalysts cool down back to RT (the return to the inactive state), which indicates that the active sites emerge under the reaction conditions exclusively. It should be mentioned that the Pd3d_5/2_ component with a higher BE, as well as the analogous component observed in the Au4f_7/2_ spectrum, which were attributed to Pd and Au atoms anchored to specific structural defects present on the HOPG surface, do not undergo variations under the conditions of carbon monoxide oxidation, meaning they are impervious to catalysis and will not be considered below.

As mentioned in the experimental section, the depth of analysis depends on the kinetic energy of the emitted photoelectrons. Namely, varying the excitation energy of the synchrotron radiation leads to a change in the photoelectron kinetic energy, and consequently to a change in the effective depth of probing. Thus, to reconstruct depth distribution profiles of distinct Pd species, their fractions measured in situ at different temperatures were drawn as a function of the photoelectron kinetic energy and depth of analysis (Figure 6). At room temperature, the Pd-CO contribution decreases with an increase in the effective depth of analysis, thereby clearly demonstrating the near-surface localization of these species for both samples. The contributions of other types of Pd species that are less abundant in freshly prepared samples under exposure to reaction mixtures at RT increase with increased apparent analysis depth. This implies their concentrations are higher in deeper-lying subsurface layers within the Pd-Au particles. When the catalyst is active (above 100–150 °C), the relative fraction of the Pd-CO-adsorbed species decreases prominently, which is a clear manifestation of progressive removal of carbon monoxide molecules from the nanoparticle surface. The desorption of CO initiated above 200 °C gives rise to a larger fraction of the alloy component, implying the incorporation of Pd atoms that were previously bonded to CO into the alloy structure. The nominal increase in the Pd-CO fraction with the depth of analysis at 250 °C could be explained by its strong overlap and resultant confusion with oxidized Pd species, as was already discussed above. After cooling the catalysts in the gas mixture back to room temperature, the initial trends of concentration profiles of Pd states are restored, which indicates that the temperature-driven modification of the surface composition of bimetallic Pd-Au particles is essentially reversible.

The results reported herein show that the activities of Pd-Au/HOPG model catalysts towards the CO oxidation are different for the two samples with different initial Pd/Au atomic ratios. Indeed, the PdAu-2 sample with a lower Pd/Au surface ratio (~0.75) is active at 100–150 °C, while the PdAu-1 with a higher Pd/Au surface ratio (~1.0) becomes active only at 200 °C and higher. Their exposure to the reaction mixture at RT induces the palladium surface segregation accompanied by an enrichment of the surface of bimetallic Pd-Au particles with palladium due to the CO adsorption on Pd atoms. The segregation extent depends on the initial Pd/Au surface ratio. The lower extent of the CO-induced segregation of Pd atoms for the PdAu-2 sample could be explained by the lower CO adsorption energy. In the active state of PdAu-1 (at 200 °C and higher), the decomposition of the Pd-CO state due to CO desorption occurs simultaneously with the PdAu surface alloy formation, while for the PdAu-2 sample, an appreciable fraction of PdAu alloy exists on the surface already at RT in the reaction mixture. Thus, it is the alloyed surface that is responsible for the activity of Pd-Au bimetallic catalysts towards the CO oxidation. Importantly, for the sample with a lower Pd/Au ratio, no formation of Pd^0^ occurs regardless of the experimental conditions, while for PdAu-1 with a high Pd/Au ratio, metallic palladium is present from the beginning, probably due to the excess of palladium in this sample. For the PdAu-1 sample under reaction mixture at RT, the Pd-Au alloy is practically absent on the surfaces of bimetallic particles due to the CO-induced segregation effect. The transformation into the alloyed surface occurs only above 150 °C. For the PdAu-2 sample, an appreciable amount of the alloyed state, which is responsible for the catalytic activity, is present on its surface even at RT, probably due to the lower CO adsorption energy or insufficient number of available palladium atoms on the surface to cover it completely and form stable Pd-CO states deactivating the catalyst. This allows us to suggest that the difference in the activity between these two catalysts is mainly determined by the presence or higher concentration of specific active Pd sites on the surface of bimetallic particles with a lower Pd/Au ratio, which is commonly referred to as the ensemble effect. The reverse redistribution of the surface composition observed after cooling in the gas mixture back to room temperature returns the catalysts into the inactive state, which strongly implies that the specific sites providing maximum activity emerge under reaction conditions only.

The activity and selectivity of bimetallic catalysts are often discussed within the context of the ensemble (geometric) and ligand (electronic) effects [13,18,32,33,34,45,46,47,48]. The former addresses the formation of a contiguous palladium domain when diluting the Pd lattice with Au. In turn, the latter implies the charge transfer between Au and Pd atoms or atomic orbit rehybridization involving either one or both metals due to the formation of heteronuclear metal–metal bonds. Most studies so far have focused on the ensemble effects as expressed in catalytic chemical reactions, which presumes that chemical bond breakage and formation within small molecules adsorbed on the alloyed bimetallic nanoparticles are mainly determined by their surface composition and the subsequent exact geometry of polymetallic sites. The results reported herein are in line with the mechanism suggested by Goodman’s group [13,18,32], which states that Au atoms provide CO adsorption, whereas adjacent Pd atomic configurations are responsible for the dissociation of dioxygen molecules into single atoms and a further spillover of O_ads_ to Au or to isolated Pd sites immediately followed by the oxidation of the activated adsorbed CO. Thus, following this mechanism, the oxidation of carbon monoxide is expected to occur at moderately low temperatures when a weakly sufficient concentration of activated adsorbed CO molecules is reached due to interactions between CO and Au. This suggestion is further strengthened by the results reported in the literature. For example, Qian et al. studied the activity of Pd-Au/SiO_2_ catalysts towards the CO oxidation in a fixed-bed flow reactor [47]. They demonstrated that for the Au-Pd bimetallic particles characterized by a high Au/Pd ratio, the surface is dominated by isolated Pd atoms, while for the nanoparticles characterized by a low Au/Pd ratio, the surface is mainly composed on multiatomic Pd_n_ configurations. It was suggested that the Pd_n_ sites mainly catalyze the CO oxidation. Cheng et al. [48], based on combined analytic potential and first-principle DFT calculations, proposed that it is the Au_43_Pd_12_ clusters that should manifest a higher activity towards the CO oxidation, while the clusters with either higher or lower Au concentrations should be less active. Non-trivial correlations between the bimetallic Pd–Au alloy composition and catalytic activity towards the CO oxidation were suggested. Delannoy et al. [10] elucidated the carbon monoxide oxidation over Pd-Au/TiO_2_ catalysts using diffuse reflectance infrared Fourier transform spectroscopy (DRIFTS) and environmental transmission electron microscopy (ETEM) techniques. They demonstrated the occurrence of Pd segregation under the influence of the CO + O_2_ reaction mixture. In this case, Pd segregation was accompanied by a loss of activity towards the oxidation of carbon monoxide, which was explained by the authors as a replacement of Au atoms with Pd in less coordinated sites.

Thus, it seems meaningful to suggest that the optimum Au/Pd ratio encouraging the emergence of multi-atomic Pd_n_ sites in parallel with the availability of Au atoms capable of adsorbing CO in the low-temperature mode is a prerequisite to achieve the highest catalytic activity of bimetallic Pd-Au catalysts in the CO oxidation reaction. Obviously, the catalytic performance strongly correlates with the initial Au/Pd ratio, as well as with exact conditions of catalytic tests. In our opinion, such results could be used as the basis for the purposeful synthesis of highly efficient Pd single-site catalysts for potential applications in low-temperature CO oxidation at the industrial scale.

## 4. Conclusions

The combination of NAP XPS and MS was applied to study the CO oxidation over model alloyed Pd–Au/HOPG catalysts with different metal ratios (Pd/Au) on the surface. We reliably identified several types of surface species and monitored changes in their concentrations under the influence of a reaction mixture, depending on the sample temperature. The catalytic activities were found to be different for two samples with different initial Pd/Au atomic ratios. More specifically, the PdAu-2 sample with a lower Pd/Au surface ratio (~0.75) was already active at 100–150 °C, while the PdAu-1 sample with a higher Pd/Au surface ratio (~1.0) became active only at 200 °C and higher. At temperatures below 100 °C for both samples, the exposure to the mixture of carbon monoxide and oxygen gives rise to a disturbance of the alloy structure due to surface segregation of palladium atoms accumulating on the surface and depleting subsurface layers. The segregation extent depends on the initial Pd/Au surface ratio. The lower extent of CO-induced segregation of Pd atoms for the PdAu-2 sample could be explained by the lower CO adsorption energy. On the one hand, the decomposition of Pd-CO adsorption complexes due to CO desorption occurs simultaneously with the PdAu surface alloy reformation in the active state of PdAu-1 (above 150 °C). On the other hand, an appreciable fraction of Pd-Au alloy exists on the surface even at RT in the reaction mixture for the PdAu-2 sample with a low initial Pd/Au surface ratio. Thus, it most likely that is the alloyed surface that is primarily responsible for the activity of Pd-Au bimetallic catalysts towards the CO oxidation. Overall, this means that the segregation effects do not directly affect the final structure of the active sites, which are to a greater extent determined by the initial Pd/Au ratio on the surface of bimetallic nanoparticles. Nevertheless, a significant segregation of Pd atoms driven by the energetically favorable formation of Pd–CO bonds could lead to a complete inhibition of the surface in the low-temperature regime for specific sample compositions. Therefore, we suggest that the difference in the activity levels of these two catalysts is determined by the presence or concentration balance of specific active Pd sites on the surface of bimetallic particles with lower Pd/Au ratios, i.e., by the ensemble effect.

## Figures and Tables

**Figure 1 nanomaterials-11-03292-f001:**
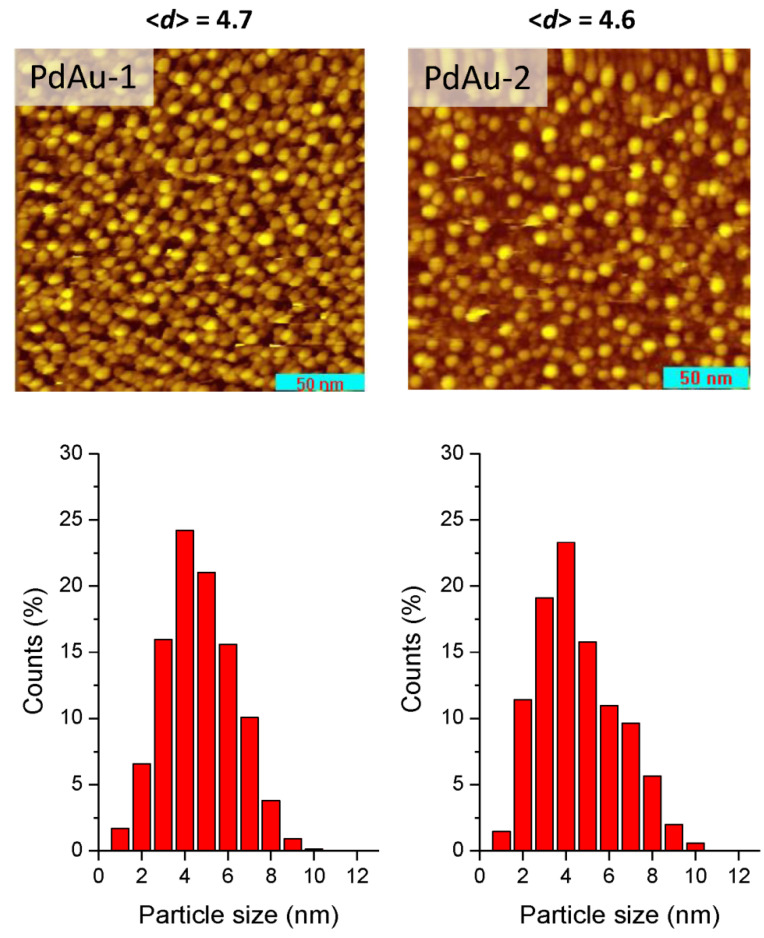
STM images (**upper panels**) and histograms of particle sizes (**bottom panels**) for the prepared model catalysts: PdAu-1 (**left**) and PdAu-2 (**right**). Tunneling parameters: 0.51 nA, −600 mV (PdAu-1); 0.56 nA, +1500 mV (PdAu-2).

**Figure 2 nanomaterials-11-03292-f002:**
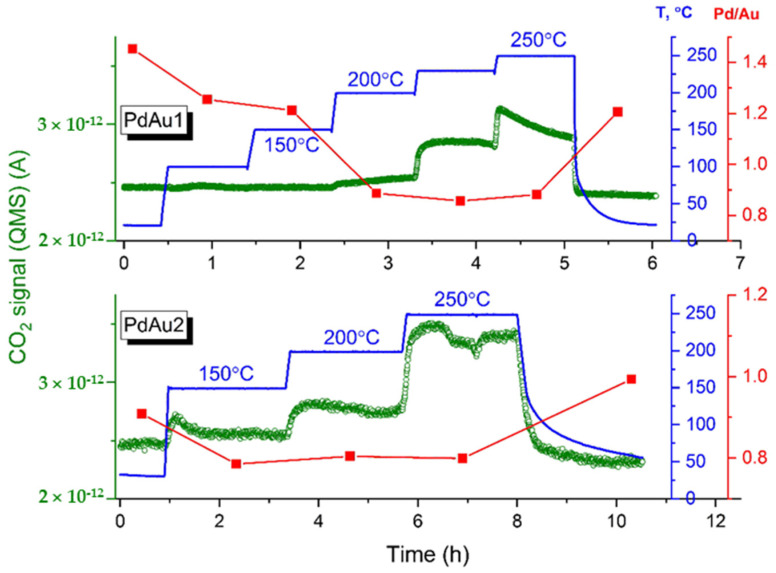
The mass-spectrometric signal of CO_2_ (corresponding to *m*/*z* = 44) (shown as green open circles) and Pd/Au atomic ratios (shown as red squares connected with a line) calculated from Pd3d and Au4f X-ray photoelectron spectra measured for PdAu-1 (**upper panel**) and PdAu-2 (**bottom panel**) samples in the CO + O_2_ reaction mixture as a function of temperature (shown as blue lines).

**Figure 3 nanomaterials-11-03292-f003:**
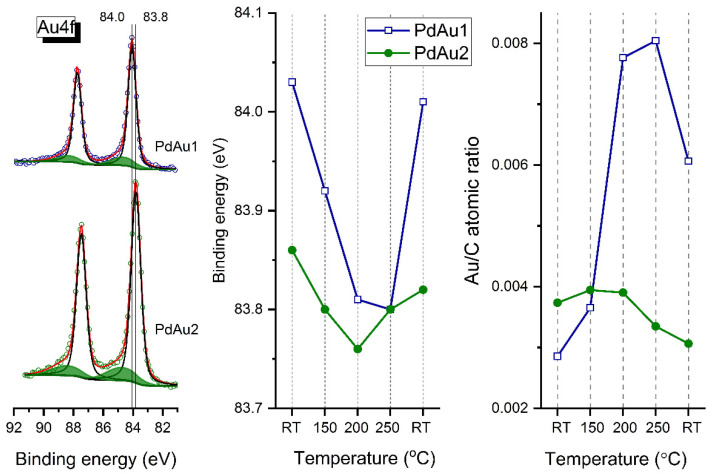
Au4f XP spectra for the PdAu-1 and PdAu-2 model alloy catalysts at room temperature (**left panel**) along with Au4f_7/2_ BEs (**central panel**) and Au-to-carbon atomic ratios (**right panel**) determined in the in situ mode in the mixture of CO and O_2_ (2:1) at a total pressure of 0.25 mbar and different temperatures. The kinetic energy of the photoelectrons was 300 eV (the depth of analysis was 1.8 nm).

**Figure 4 nanomaterials-11-03292-f004:**
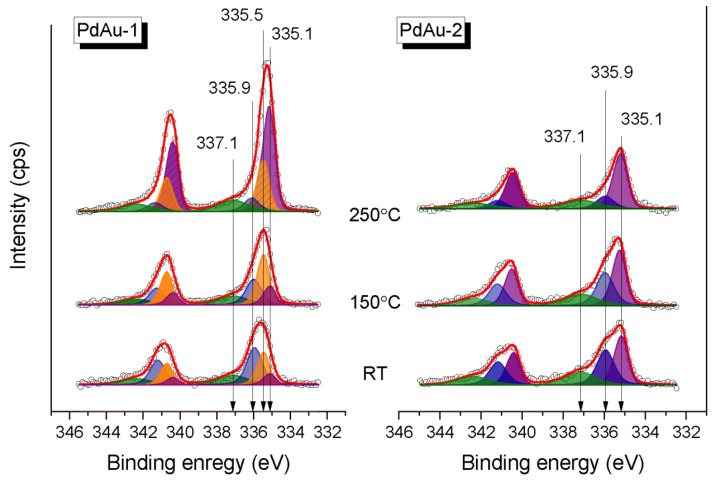
Pd3d XP spectra measured for PdAu-1 and PdAu-2 samples in the carbon monoxide: oxygen mixtures (CO/O_2_ = 2:1, P_total_ = 0.25 mbar) at RT, 150 °C and 250 °C. The kinetic energy of photoelectrons was 300 eV (the depth of analysis was 1.8 nm).

**Figure 5 nanomaterials-11-03292-f005:**
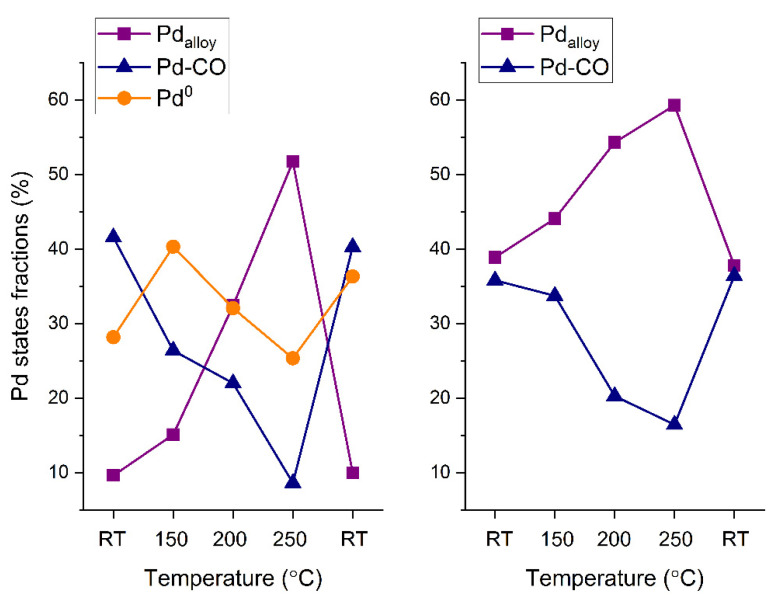
Fractions of different Pd states for the PdAu-1 (**left**) and PdAu-2 (**right**) samples in the reaction mixture (CO/O_2_ = 2:1) as a function of temperature (calculated from XP spectra measured at the kinetic energy level of 300 eV).

**Figure 6 nanomaterials-11-03292-f006:**
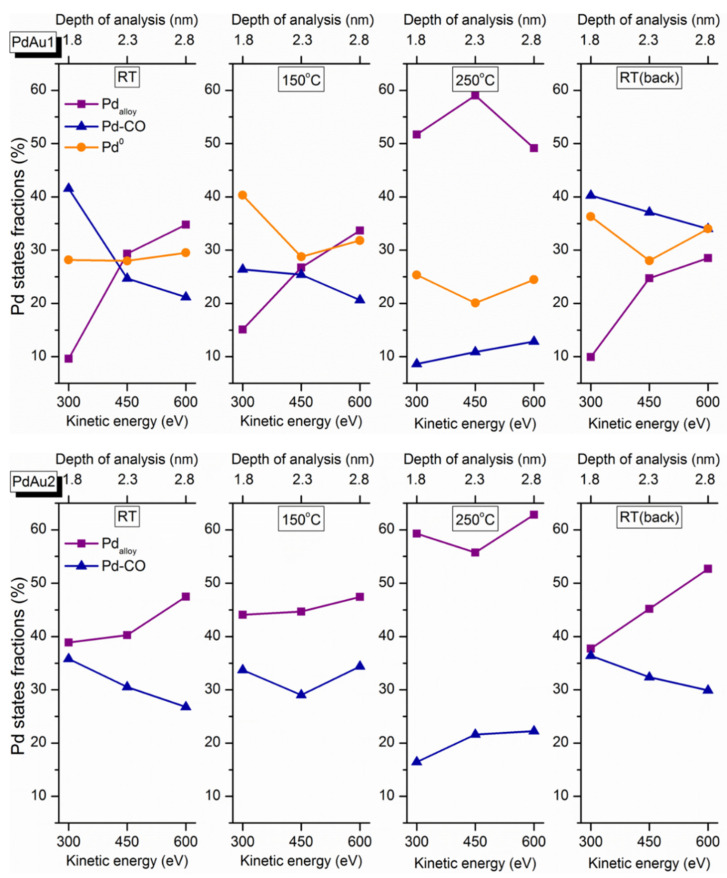
Depth profiles of fractions corresponding to distinct Pd states for different temperatures for PdAu-1 (**upper panels**) and PdAu-2 (**bottom panels**).

**Table 1 nanomaterials-11-03292-t001:** Element ratios calculated from the XPS data measured in UHV after annealing at 400 °C (photoelectron kinetic energy of 300 eV corresponding to the surface localization).

Sample	Au/C	Pd/C	Pd/Au
PdAu-1	0.003	0.003	1.0
PdAu-2	0.004	0.003	0.75

## Data Availability

The datasets generated and analysed during the current study, including raw spectra and images, are available from the corresponding author on reasonable request.

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
