# Peer review of "Near-Ambient Pressure XPS and MS Study of CO Oxidation over Model Pd-Au/HOPG Catalysts: The Effect of the Metal Ratio"

_nanomaterials, 2021, doi:10.3390/nano11123292_

Round 1
Reviewer 1 Report
Nanoparticles supported on carbon are a technological catalyst of interest. There are plenty of studies available, dealing with HOPG supported model catalysts. Important pioneering work is missing in the list of citations:
1) M. Favaro et al, PhysChemChemPhys 15(2013) 2923
2) A. Motin et al, ApplSurfSci 440(2018) 680
The authors must not leave these papers unmentioned!
The weakest point in the analytical secion, where the Au-Pd composition is analysed, deals with the superposition of the Au4d5 and the Pd3d peaks in XPS. It is crucial, that the raw XPS spectra as well as the spectra aufer Au4d5 subtraction are shown. A recent study dealing with exactry this problem can be found in:
3) A. Teixeira-Neto et al, CatSciTec 8(2017) 1679.
This should be cited, too.
Author Response
Nanoparticles supported on carbon are a technological catalyst of interest. There are plenty of studies available, dealing with HOPG supported model catalysts. Important pioneering work is missing in the list of citations:
1) M. Favaro et al, PhysChemChemPhys 15(2013) 2923
2) A. Motin et al, ApplSurfSci 440(2018) 680
The authors must not leave these papers unmentioned!
We thank the reviewer for pointing this out. We have cited these additional references in the Introduction section of the Manuscript.
The weakest point in the analytical secion, where the Au-Pd composition is analysed, deals with the superposition of the Au4d5 and the Pd3d peaks in XPS. It is crucial, that the raw XPS spectra as well as the spectra aufer Au4d5 subtraction are shown. A recent study dealing with exactry this problem can be found in:
3) A. Teixeira-Neto et al, CatSciTec 8(2017) 1679.
This should be cited, too.
Many thanks for noting this as well. Actually, there are two different methods to subtract the Au4d5/2 component from measured XP spectra. In the work by A. Teixeira-Neto et al. (CatSciTec, 8 (2017), 1679), the authors did not measure Au4d3/2 and used the parameters of Pd3d3/2 to fit Pd3d5/2; the residual peak was ascribed to Au4d5/2. In our work, we measured the longer binding energy region, i.e., the entire Pd3d and Au4d lines (encompassing both 5/2 and 3/2 lines). The Au4d3/2 line does not overlap with Pd3d and could be used to fit Au4d5/2, which strongly overlaps with the Pd3d region. Knowing theoretical value of 2:3 for the branching ratio of the spin-orbit doublet and parameters of the Au4d3/2 component make it possible to fit Au4d5/2 and then subtract it from the raw XPS data to get undisturbed Pd3d region. Both methods make sense to be used. We have cited this work in the manuscript and added the following information in the Experimental part:
There are two possible ways to take into account the Au4d5/2 contribution correctly. The first one is described in ref. [30], where best-fit parameters of the Pd3d3/2 compo-nent are applied to fit the Pd3d5/2 component using the theoretical spin-orbit splitting. The resultant residual intensity is thus ascribed to the Au4d5/2 line. The second way is based on fitting the Au4d5/2 component with parameters taken from the Au4d3/2 line, which does not overlap with the Pd3d line. Both these ways make sense and give quite similar results. In this work, the second one was used, i.e., the shape and intensity of the Au4d5/2 line were calculated from the less intense Au4d3/2 peak using the theoretical value of 2:3 for the branching ratio of the spin-orbit doublet components.

Reviewer 2 Report
This paper uses the combination of NAP XPS and MS methods to study the relationship between the catalytic activity of bimetallic Pd-Au/HOPG catalysts for CO oxidation and the atomic ratio of Pd/Au. The authors found that the catalytic activities of two samples with different initial Pd/Au atomic ratios were different. Overall, it is an interesting study. The test and analysis procedures are detailed and reliable. The manuscript can be accepted with the following revisions:
- The author is kindly requested to provide the CO conversion rate diagram with different Pd/Au ratios, which can provide more intuitive reference value for industrial production.
- Page 3, Line 2: “different mean particle sizes and Pd/Au ratios were prepared by PVD.” Is PVD an acronym? The author should give an accurate expression of it.
- Page 6, Line 1: “These results are in accord with the Goodman’s mechanism, which assumes that CO oxidation should occur at a relatively low temperature when weak interactions between CO and Au can provide a sufficient concentration of adsorbed CO molecules.” Why?
- Page 9: Figure 6 (in order to obtain information about the depth distributions of different Pd states), the author should explain it more deeply and fully.
Author Response
This paper uses the combination of NAP XPS and MS methods to study the relationship between the catalytic activity of bimetallic Pd-Au/HOPG catalysts for CO oxidation and the atomic ratio of Pd/Au. The authors found that the catalytic activities of two samples with different initial Pd/Au atomic ratios were different. Overall, it is an interesting study. The test and analysis procedures are detailed and reliable. The manuscript can be accepted with the following revisions:
The author is kindly requested to provide the CO conversion rate diagram with different Pd/Au ratios, which can provide more intuitive reference value for industrial production.
In general, we agree with the reviewer that this issue is important. First of all, we would like to mention that we work with model catalysts with metal nanoparticles (PdAu) deposited onto a planar support (HOPG). The use of such model catalysts could help to get more reliable data concerning the surface structure and chemical composition of active metals depending on different treatments since in the case of “real” industrially applicable catalysts, where nanoparticles of active metals are deposited onto a high-specific-surface area support, the application of surface-sensitive techniques is limited due to low surface concentration of the active component. From the other side, the total amount of active sites in model systems is much lower as compared to the “real” catalysts since the metal nanoparticles nominally form a monolayer atom the atomically smooth support. Furthermore, catalytic tests on model systems under the conditions, which are normally used for real catalysts, give effectively negligible conversion/products yields. Thus, there are no meaningful reasons to compare the catalytic properties of model catalysts with those obtained for real catalysts. The current study was aimed at deeper understanding of factors governing the catalytic activity dependence on the metals ratio rather than to develop a new type of efficient catalysts.
Nevertheless, we indeed estimated the conversion of CO from mass spectrometry data. It did not exceed 10% for all measurements. From the MS data presented in Fig. 2, one can see that the samples show comparable increases in the mass-spectrometric CO2 signal at 250°C. But it should be mentioned that this difference should be also corrected by the number of active sites on the surface of model catalysts. In the case of monometallic Pd/HOPG model catalyst it could be estimated from the STM (particle size distribution) and XPS data (atomic concentration of Pd). But in the case of bimetallic PdAu/HOPG model catalysts, first of all, we need to determine the structure of the active sites, which are responsible for the activity. In our opinion, it is more important that the activities of the two model catalysts differing from each other in the initial Pd/Au atomic ratios appeared distinctly different in terms of ignition temperature.
Page 3, Line 2: “different mean particle sizes and Pd/Au ratios were prepared by PVD.” Is PVD an acronym? The author should give an accurate expression of it.
Thanks for pointing this out. PVD (physical vapor deposition) was substituted by thermal vacuum deposition in the text.
Page 6, Line 1: “These results are in accord with the Goodman’s mechanism, which assumes that CO oxidation should occur at a relatively low temperature when weak interactions between CO and Au can provide a sufficient concentration of adsorbed CO molecules.” Why?
We actually mean that the shift of the activity ignition temperature to lower values in the case of the sample with a lower Pd/Au surface atomic ratio is consistent with the assumptions made in works published by Prof. Goodman's group. According to this, the CO oxidation should occur at a relatively low temperature when weak interactions between CO and Au can provide a sufficient concentration of adsorbed CO molecules. The text of the manuscript has been corrected accordingly.
Page 9: Figure 6 (in order to obtain information about the depth distributions of different Pd states), the author should explain it more deeply and fully.
Thanks for pointing this out. We have tried to extend discussion in the text of the manuscript.

Reviewer 3 Report
Title “Near Ambient Pressure XPS and MS Study of CO Oxidation over Model Pd-Au/HOPG Catalysts: The Effect of Metal Ratio”.
It is interesting that the paper presented the the effect of PdAu ratio on the CO oxidation with XPS and MS in an ambient pressure. The authors found that the optimum active sites emerge only under reaction conditions including both elevated temperature and reactive atmosphere. And the different activity of the two catalysts is attributed to a ensemble effect, which is related to the the presence and/or concentration balance of specific Pd active sites on the PdAu surface.
However, the paper can be acceptable before the authors provide the following information.
- The authors should also provide the detailed synthesis method and the mechanism for the two different catalysts, since we could not found any information about the reason for the two different catalysts with the same method in the similar prepared condition.
- In Fig. 1, I could not distinguish the morphology and the size of the nanoparticles. And the average size of PdAu-2 looks like much bigger than that of PdAu-1, TEM should also be provided. Furthermore, HRTEM should be provided for the illustration of the hemispherical and/or truncated hemispherical shapes for the nanoparticles.
- For the quantitative analysis, the data with ICP-OES should also be provided, since we know that XPS is just useful for surface analysis and is not very accurate for the quantitative analysis. Even this paper is focused on the surface analysis, the content of the bulk PdAu is also determined the surface ratio of the Pd:Au.
Author Response
The authors should also provide the detailed synthesis method and the mechanism for the two different catalysts, since we could not found any information about the reason for the two different catalysts with the same method in the similar prepared condition.
Thank you for pointing this out. We agree that the preparation procedure of the samples is described in insufficient detail. We have specially prepared two samples with different Pd/Au atomic ratios using consecutive thermal vacuum metal deposition procedure. We have added information about deposition conditions, i.e., about accelerating voltage and thermo-emission current for both metals. It is worth noting that the amount of metal deposited was varied by the duration of deposition and controlled by XPS. Thus, such a procedure allows us to prepare HOPG-supported catalysts with different Pd/Au atomic ratios on the surface but similar or presumably identical other essential characteristics. The following information was added to the text of manuscript.
Deposition conditions were as follows: accelerating voltage was ~900 V for Pd and 800 V for Au, the thermoemission current was ~15 mA for both metals. The amount of the metal deposited was varied by the duration of the deposition and was controlled with XPS.
In Fig. 1, I could not distinguish the morphology and the size of the nanoparticles. And the average size of PdAu-2 looks like much bigger than that of PdAu-1, TEM should also be provided. Furthermore, HRTEM should be provided for the illustration of the hemispherical and/or truncated hemispherical shapes for the nanoparticles.
Thanks for noting this. First of all, it is worth mentioning that HOPG single crystals (square plates 7×7 mm, approx. 1 mm thickness) were used as supports. And metal nanoparticles were deposited onto their surface to form a nominal monolayer. Thus, if we want to apply TEM and/or HRTEM for studying these samples we have to cleave their topmost parts (with a thickness of only a few graphene layers), which would decisively destroy the sample. Moreover, we stress that the PdAu/HOPG samples are deliberately designed for studies using surface-sensitive techniques, especially NAP XPS. Due to this reason, for the surface morphology investigation, we applied non-destructive surface sensitive technique, STM, which is perfectly appropriate for this purpose. As concerns the sizes of the presented nanoparticles, they may not be compared with each other directly «by a naked eye» due to different contrast of the images, which could form a fake impression that one particle is bigger that another one, while they are actually of the same size. Due to this reason, we calculated the mean particle sizes and histograms of particle size distribution. The mean particles diameter was determined from the STM images by the ParticlesNN Web-based application based on nanoparticle shape recognition with machine learning algorithms as it mentioned in the Experimental part (for more details, see refs. 22-24 in the revised manuscript). Additionally, the STM images for these samples were processed and analyzed using the XPMPro 2.0 control software package (the measurements were done for each particle using manual precise profile analysis). The results obtained by both methods were essentially identical. It should be also noted that when STM data processing is performed manually the size of the particle is determined as it is presented in the Figure below. At the same time, the algorithm implemented in the ParticlesNN Web-based application is similar.
The issue of exact shape of nanoparticles was considered in detail in our recently published paper devoted to the evaluation of gold particle sizes using XPS data (M. Y. Smirnov, A. V. Kalinkin, A. V. Bukhtiyarov, I. P. Prosvirin, V. I. Bukhtiyarov, Using X-ray photoelectron spectroscopy to evaluate size of metal nanoparticles in the model Au/C samples, J. Phys. Chem. C. 120 (2016) 10419–10426. https://doi.org/10.1021/acs.jpcc.6b02090 – ref. 28). In particular, we found that in the case of pristine HOPG sample not treated with argon ions, the deposited gold particles had a predominantly spherical shape due to the fact that the interaction between the gold particles and carbon atoms on the flat surface of graphite was weak. However, the particle size distribution in that case was strongly inhomogeneous, and the particles were mostly concentrated at the support steps. To achieve a homogeneous particle size distribution, an additional synthesis stage was proposed implying a short treatment of the pristine HOPG surface with argon ions leading to the stabilization of nanoparticles at defects intentionally introduced by the treatment. Such a modification of the HOPG surface with the subsequent deposition of gold resulted in the formation of particles with hemispherical and truncated hemispherical shapes. We used basically the same deposition protocol in our work, so we suppose that the bimetallic Pd-Au nanoparticles deposited onto the modified HOPG surface also have essentially hemispherical and truncated hemispherical shapes. We extended the discussion about this in the manuscript.
For the quantitative analysis, the data with ICP-OES should also be provided, since we know that XPS is just useful for surface analysis and is not very accurate for the quantitative analysis. Even this paper is focused on the surface analysis, the content of the bulk PdAu is also determined the surface ratio of the Pd:Au.
First of all, we would like to mention that model catalysts with metal particles (PdAu) deposited onto a planar support (HOPG) were used in this study. The use of such model catalysts could help to get more reliable data concerning the surface structure and chemical composition of active metals depending on different treatments since in the case of real catalysts composed of nanoparticles of active metals deposited onto a high specific-surface-area support, the application of surface-sensitive techniques is limited due to low surface concentration of the active component.
An application of the ICP-OES analysis for such samples is challenging due to the very low apparent concentration of the metals (the nanoparticles form a monolayer atop the atomically smooth support. Anyway, this technique is destructive for the sample (it requires its dissolution). Thus, it would not possible to continue to work with the sample subjected to ICP-OES analysis. Thus, it is hardly probable that the ICP-OES technique would provide any meaningful and useful information for such model planar systems.
Concerning the total amounts of each metal deposited and average composition of the systems, we fully agree with the reviewer that the use of an independent integral method, which could quantitatively provide a total amount of the metal deposited, would be very helpful for our study. Unfortunately, we did not have the possibility to measure the amounts of deposited metals independently. The amount was controlled just using XPS measurements. But, in principle, it is not really important in our case due to the fact that we do not really use the total amount of metals in order to derive some scientific conclusions, for example, catalytic activity calculations, etc.

Round 2
Reviewer 3 Report
The authors improved the expression in the revised paper. However, for the sizes comparison of the nanoparticles, we have to calculate at least 50 particles with some tools instead of a naked eye. But the direct observation with our naked eyes is really related to what we obtained with any tools. At least, the two pictures are not suitable here. It is better to provide more STM pictures to prove your conclusions.
Author Response
Indeed the naked eyes usually work pretty well for the STM images. We would like to emphasize that in all cases we performed a series of STM measurements at several different locations on the sample surface using varied STM data acquisition parameters (namely, current, bias voltage, tip scanning rate, etc.) to check for reproducibility of the particle shapes and size distributions. The histogram of particle size distribution was based on the measurements of at least 1000 particles for each sample. So we attach the Figure with the additional STM images to the reply. Finally we basically changed the STM image for the PdAu-1 to more representative/reliable in the Figure 1.
